# The Acute Effects of an Ultramarathon on Atrial Function and Supraventricular Arrhythmias in Master Athletes

**DOI:** 10.3390/jcm11030528

**Published:** 2022-01-20

**Authors:** Luna Cavigli, Alessandro Zorzi, Veronica Spadotto, Giulia Elena Mandoli, Andrea Melani, Chiara Fusi, Antonello D’Andrea, Marta Focardi, Serafina Valente, Matteo Cameli, Marco Bonifazi, Flavio D’Ascenzi

**Affiliations:** 1Department of Medical Biotechnologies, Division of Cardiology, University of Siena, 53100 Siena, Italy; 2Department of Cardiac, Thoracic and Vascular Sciences and Public Health, University of Padova, 35131 Padova, Italy; 3Ospedale Riabilitativo di Alta Specializzazione di Motta di Livenza, 31045 Treviso, Italy; 4Unit of Cardiology and Intensive Coronary Care, “Umberto I” Hospital, 84014 Nocera Inferiore, Italy; 5Department of Medicine, Surgery and Neuroscience, University of Siena, 53100 Siena, Italy

**Keywords:** endurance, right atrium, left atrium, athlete’s heart, master athletes, arrhythmias, speckle-tracking echocardiography, sports

## Abstract

Background. Endurance sports practice has significantly increased over the last decades, with a growing proportion of master athletes. However, concerns exist regarding the potential proarrhythmic effects induced by ultra-endurance sports. This study aimed to analyse the acute effects of an ultramarathon race on atrial remodelling and supraventricular arrhythmias in a population of master athletes. Methods. Master athletes participating in an ultramarathon (50 km, 600 m of elevation gain) with no history of heart disease were recruited. A single-lead ECG was recorded continuously from the day before to the end of the race. Echocardiography and 12-lead resting ECG were performed before and immediately at the end of the race. Results. The study sample consisted of 68 healthy non-professional master athletes. Compared with baseline, P wave voltage was higher after the race (*p* < 0.0001), and more athletes developed ECG criteria for right atrial enlargement (*p* < 0.0001). Most of the athletes (97%) had ≥1 premature atrial beats (PAB) during the 24-h monitoring, also organised in triplets (17%) and non-sustained supraventricular tachycardias (NSSVTs) (19%). In contrast, exercise-induced PABs, triplets, and NSSVTs were rare. One athlete developed acute atrial fibrillation during the race. After the race, no significant differences were found in biatrial dimensions. Biatrial function, estimated by peak atrial longitudinal and contraction strains, were normal both before and after the race. Conclusions. In master athletes running an ultramarathon, acute exercise-induced atrial dysfunction was not detected, and exercise-induced supraventricular arrhythmias were uncommon. These results did not confirm the hypothesis of an acute atrial dysfunction induced by ultra-endurance exercise.

## 1. Introduction

Moderate exercise has well-recognised favourable effects on the cardiovascular system, but there is an incomplete understanding of the entire dose–response relationship [1,2,3]. Similarly to a drug, an insufficient dose may not confer the optimal benefits while an excessive amount may cause harm [4]. Endurance events, including ultramarathons and long-distance races, have gained increasing popularity. The growing popularity of endurance and ultra-endurance sports events has been paralleled by an increasing number of participants ≥40 years old (so-called “master athletes”), who may be more prone to cumulative cardiac damage [5,6]. Notably, endurance athletes often exercise 15–20 times more than recommended by current guidelines [7,8].

Middle-aged endurance athletes have an increased risk of developing atrial fibrillation (AF) and atrial flutter [9] and a dose–response relationship between endurance training load and AF risk has been demonstrated [10]. However, the mechanisms linking AF and endurance training are not entirely clear [11]. Although atrial dilatation is typical of the athlete’s heart, the atrial contractility remains normal and such physiological changes are unlikely to represent a substrate for the occurrence of supraventricular arrhythmias [12,13,14,15]. High-intensity exercise sessions may cause a transient rise in cardiac and inflammatory biomarkers, acute inflammation, increased atrial wall tension, and atrial dysfunction [16,17]. Although these modifications are reversible, it has been postulated that atrial damage may develop after years of intense physical exercise, generating a substrate for atrial arrhythmias [16,17]. Furthermore, the acute effects of ultra-endurance races in master athletes have been rarely explored with a contemporary evaluation based on advanced echocardiography and ECG monitoring before and during the race.

To further explore the relationship between strenuous exercise and the atria, we evaluated the acute exercise-induced effects of an ultramarathon race on atrial function by standard and advanced echocardiography and assessed the occurrence of ECG changes and supraventricular arrhythmias before, during, and immediately after the race in a sample of non-professional master athletes.

## 2. Methods

This study was conducted during the 2020 “Terre di Siena Ultramarathon” (Siena, Italy), which started in San Gimignano and finished in Siena (50 km, 600 m of elevation gain). The ultramarathon is defined as a race with a distance greater than the official marathon (42,195 m) and may vary according to a certain range or time limit (i.e., 50 km, 100 km, and longer events or 6 h, 24 h, and multi-day events) [18] (see also https://www.worldathletics.org/disciplines/ultra-running/ultra-running, last accessed on 1 December 2021). Athletes were recruited on a voluntary basis and signed informed consent. All subjects gave their informed consent for inclusion before they participated in this study. This study was conducted in accordance with the Declaration of Helsinki, and the protocol was approved by the local Ethics Committee (vers. 1.0/20 December 2019).

### 2.1. Study Group

Non-professional athletes ≥40 years old participating in the “Terre di Siena Ultramarathon” were offered participation. According to the Italian law, all athletes must undergo annual pre-participation evaluation, based on physical examination, 12-lead resting, and exercise ECG, to be considered eligible for sports competition. All athletes participating in this study were considered eligible for competitions. Athletes who withdrew before the finish line, arrived over the maximum time, or refused to repeat the investigations after the race were excluded. A total of 71 athletes were initially enrolled: 1 athlete was excluded for withdrawing before the finish line, 1 for arrival at the finish line over the maximum time, and 1 because of the onset of AF with a high heart rate after the beginning of the race. Given the aim of this study, the pre-race data of the athlete experiencing AF were reported separately in the results paragraph. The final population consisted of 68 healthy non-professional master athletes.

### 2.2. Electrocardiography

All participants underwent a 12-lead resting ECG, recorded at a speed of 25 mm/s and standardised calibration for 10 mm/cm. Recordings were performed using CARDIOLINE 200 S 12-lead ECGs (Cardioline S.P.A.) the day before the marathon and immediately after the race, at the finish line. ECG tracings were reviewed independently by two experienced cardiologists (A.Z. and F.D.), and consensus solved discrepancies. Resting heart rate (HR) and PR interval were evaluated in accordance with the international criteria [19]; P-wave amplitude and P-wave duration were measured in lead II to identify left atrium enlargement (LAE) and right atrium enlargement (RAE). LAE was defined as P-wave duration > 120 ms in lead I or II with negative portion of the P-wave of ≥0.1 mV in depth and ≥40 ms in duration in the lead V₁. RAE was defined as the P-wave amplitude ≥ 0.25 mV in lead II [19].

### 2.3. ECG Monitoring

A single-lead ECG monitoring of 24–30 h was performed in all athletes using the RootiRx^®®^ device (Rooti Labs Ltd., Taipei, Taiwan). RootiRx^®®^ is a small device consisting of an integrated multisensor system, a microelectronic board with memory storage, and an internal rechargeable battery. It has a Conformité Européenne (CE) mark and Food and Drug Administration (FDA) clearance. The device has already been described in detail in our previous study [20]. Recordings were started the day before the race and interrupted immediately at the end of the ultramarathon. Two experienced cardiologists (A.Z. and F.D.) reviewed them independently, and discrepancies were solved by consensus. The recordings were analysed and reported separately for data obtained at rest and during the competition. Athletes with >50% of the recording time during the race not suitable for interpretation because of artefacts were excluded from the analysis.

### 2.4. Echocardiography

Echocardiographic examination was performed using two high-quality portable echocardiographs (Vivid iq, GE, Milwaukee, WI, USA), equipped with an M4S 1.5-MHz to 4.0 MHz transducer, and a one-lead ECG was continuously displayed. Recordings were performed the day before the marathon and immediately after the race, at the finish line. Subjects were studied in the steep left-lateral decubitus position. Two experienced readers (A. M. and C. F.) performed offline data analysis using dedicated software (EchoPac, GE, Milwaukee, WI, USA). Left atrial (LA) and right atrial (RA) dimensions were measured at the end of ventricular systole when these chambers reached their maximum size during the cardiac cycle, as recommended [21,22]. RA and LA area and volume were calculated by the biplane method of disks (modified Simpson’s rule) in the apical 4-chamber view for the former and both 4- and 2-chamber views for the latter, obtaining an average value. LA size was assessed, excluding the pulmonary veins and LA appendage from LA tracing. The mitral annulus plane was used as the inferior border [21,22]. RA dimension was measured excluding the area between the leaflets and annulus, following the RA endocardium, excluding the inferior and superior vena cava and RA appendage [23]. Biatrial volumes were indexed to body surface area (BSA), calculated using the Dubois formula [24].

#### Two-Dimensional Speckle-Tracking Echocardiography (STE)

Two-dimensional STE was obtained and recorded using conventional 2D grey-scale echocardiography with stable electrocardiographic tracing during breath-holding. All speckle tracking data were analysed offline by two experienced readers (A.M. and C.F.) using dedicated automated software (EchoPAC PC, Version 112; GE Health Care, Milwaukee, WI, USA). Endocardial surfaces were manually traced in both 4- and 2-chamber views for the left atrium by a point-and-click approach to create a region of interest with the automatic adjustment of the system. After a manual adjustment of the tracked area, the software divides the ROI into six segments, and the resulting tracking quality for each segment was automatically scored as either acceptable or non-acceptable, with the possibility for further manual correction [25,26]. Peak atrial longitudinal strain (PALS) and peak atrial contraction strain (PACS), measures of the atrial reservoir and active function, respectively, were obtained calculating the average of all segments and the average values obtained from the 4-chamber and 2-chamber view (LA global PALS and global PACS) [25,26].

### 2.5. Statistical Analysis

The normal distribution of all continuous variables was examined using the Shapiro–Wilk test. Categorical variables are expressed as percentages. According to the data distribution, the paired *t*-test and the Wilcoxon matched-paired test were used to assess the within-group significance (pre-race vs. post-race significance). The McNemar test was used for nominal data. A *p*-value < 0.05 was considered statistically significant. As appropriate for data distribution, correlation analysis was performed to find an association between the ECG and demographics or echocardiographic variables using the Spearman and Pearson methods. Statistics were performed using SPSS, version 21.0 (Statistical Package for the Social Sciences Inc., Chicago, IL, USA).

## 3. Results

The demographic characteristics of the study population are reported in Table 1. All 68 participants (mean age 47.9 ± 7.8 years) completed the ultramarathon covering 50 km (average race time: 5.5 h). The comparison between pre-race and post-race ECG data is reported in Table 2. Before the race, RAE and LAE were found in 7.4% and 32.4% of the athletes, respectively. Compared with baseline data, P-wave voltage was higher after the race (*p* < 0.0001), and more subjects developed RAE (*p* < 0.0001). No significant differences were found in PR interval, P wave duration, or LAE. Premature atrial beats (PABs) were not recorded in any 12-lead resting ECG.

### 3.1. ECG Monitoring before and during the Race

Of 68 athletes, 9 were excluded from the analysis because of artefacts that prevented interpretation of the ECG recording for more than half of the racing time. ECG monitoring data obtained before and during the entire duration of the race are reported in Table 3. During the race, athletes reached a peak heart rate of 98% (91–105) of the maximal theoreticalHR. At rest, most of the athletes had at least 1 PAB during the 24 h and couplets, triplets, and non-sustained supraventricular tachycardias were demonstrated in a sizeable proportion. Conversely, exercise-induced PABs (i.e., the occurrence of PABs only during exercise or an increase in the burden of PABs/hour during exercise) were rare. None of the athletes showed acute exercise-induced supraventricular regular tachycardia. An athlete developed symptomatic acute hemodynamically stable AF after the beginning of the race. Because of the high HR, he could not complete the race and was transported to the emergency room. Spontaneous cardioversion to sinus rhythm was demonstrated 2 h after the admission to the hospital.

### 3.2. Echocardiography

LA and RA echocardiographic data collected before and after the race are reported in Table 4. Both LA volume and LA volume index did not significantly differ before and after the race (*p* = 0.15). After the race, LA PACS values were significantly higher than baseline data (*p* < 0.001) while no differences were found for LA PALS. After the race, no significant differences were found in RA volume and RA volume indexed. RA function, estimated by RA PALS and PACS, did not significantly differ before and after the race. Although the athlete experiencing AF during the race was excluded from the final analysis, we estimated biatrial size and function in this 50-year-old male non-professional athlete: biatrial size (LA volume index: 24 mL/m^2^; RA volume index: 21 mL/m^2^) and biatrial reservoir function (LA PALS: 36%; RA PALS: 40%) were normal before the race, when the athlete was in sinus rhythm.

### 3.3. Correlation Analysis

In the correlation analysis, LA volume showed a moderate correlation with RA volume (r = 0.57, *p* < 0.001) and a weak correlation with left ventricular (LV) end-diastolic diameter (R = 0.48, *p* < 0.001), LV end-diastolic volume (R = 0.45, *p* < 0.001), and right ventricular (RV) area (R = 0.37, *p* = 0.002). RA volume weakly correlated with LV end-diastolic diameter (R = 0.45, *p* < 0.001) and with RV end-diastolic diameter (R = 0.33, *p* = 0.008).

Figure 1 summarizes the main results of the study.

## 4. Discussion

In this study, we analysed the acute effects of an ultramarathon on the electrical and mechanical function of the left and right atrium in master athletes (≥40 years old). We performed a 12-lead resting ECG and standard and advanced echocardiography before and immediately after the race to evaluate the acute exercise-induced effects during the race. Moreover, we recorded a single-lead ECG the day before the ultramarathon and during the entire duration of the competition. The main findings of this study are: (i) biatrial size was not significantly affected by the ultramarathon; (ii) biatrial function was not influenced by ultra-endurance running, as also demonstrated by advanced imaging, with normal reservoir and contractile function demonstrated before and after the race; and (iii) the vast majority of master athletes showed supraventricular arrhythmias, isolated or complex, at rest, but supraventricular arrhythmias were rarely found during the race, although an athlete developed atrial fibrillation.

### 4.1. Atrial Volumes

Biatrial remodelling has been extensively investigated in athletes, with biatrial dilatation being identified as one of the features of the athlete’s heart [27,28]. Indeed, in this study, we found a correlation among the size of the different cardiac chambers, confirming that the athlete’s heart is characterised by a harmonic and consistent increase in the dimension of all cardiac chambers while a non-harmonic (disproportionate) remodelling potentially suggests a non-physiologic process [29]. More recently, some authors have investigated atrial remodelling during exercise. Some authors found that, during endurance exercise, the volume overload causes an acute increase in atrial dimensions, and particularly in RA size [30]. Notably, in healthy middle-aged athletes, marathon running may cause RA dilation in the absence of changes in LA volume immediately after the race [31]. Wright et al. described in well-trained healthy middle-aged endurance athletes a dynamic adaptation of LA phasic volumes during light- and moderate-intensity exercise: LA maximal and reservoir volume increased from rest to light exercise in relation to atrioventricular plane displacement, without further change during moderate exercise [32]. LA passive emptying volume increased during light exercise and then returned to baseline during moderate exercise, whereas LA active emptying increased appreciably only during moderate exercise so that the total atrial emptying volume did not increase beyond light exercise [32]. This complex adaptation could be explained in part by the possibility of maintaining better LV filling in these conditions according to the Frank–Starling mechanism [32]. We did not find significant acute changes in either right or left atrial size in this study. In agreement with our results, Wilhelm et al. observed no significant differences over time in left and right atrial volumes in a longitudinal study on elite athletes running a marathon [17].

### 4.2. Atrial Function

Analysis of the biatrial remodelling in the athlete’s heart that goes beyond biatrial dimensions, and the advanced echocardiographic techniques, particularly speckle-tracking echocardiography, gave us the unique opportunity to analyse in detail biatrial myocardial deformation [33]. Although biatrial remodelling is usually associated in athletes with a normal biatrial function at rest [12,34], atrial function could be acutely affected by exercise, with impairment of atrial systolic function in endurance athletes [16]. Some evidence suggests that the physiological demands of maintaining a high cardiac workload for a prolonged time may result in a transient cardiac dysfunction, i.e., a form of exercise-induced ‘cardiac fatigue’ [35]. Sanz-de la Garza et al. analysed healthy adults at baseline and after a three-stage trail race: an acute impairment of atrial function was observed, with RA reservoir function decreasing in medium-distance runners and to a greater extent in long-distance runners, confirming a dose–response relationship between exercise load and deterioration in the performance of the atria [16]. In the RA, and less pronounced in the LA, three patterns of atrial response to exercise were documented, depending on the amount of exercise performed: an increase in both reservoir and contractile function in athletes running 14 km, a decrease in reservoir function and an increase in contractile function in athletes running 35 km, and a decrease in both reservoir and contractile function in 55-km runners [16]. However, high interindividual variability was documented, supporting the potential role of a different response to different stimuli of acute exercise in each individual [16]. The presence of interindividual variability was partly confirmed by Gabrielli et al., who observed lower LA and RA contraction strain and strain rate parameters in endurance athletes during peak exercise than non-athletes [36]. However, excluding the athletes presenting significant atrial enlargement, no significant differences were noticed between athletes and sedentary controls, suggesting that a subgroup of athletes with greater atrial size shows lower atrial deformation during exercise [36]. Chen et al. analysed biatrial function in triathletes after an endurance race using novel feature-tracking cardiac magnetic resonance (CMR): they demonstrated that LA global longitudinal strain (GLS) decreased post-race while RA GLS remained constant, accompanied by a decrease in biatrial volumes [37]. In our study, in non-professional master athletes running an ultramarathon, no differences were found in LA PALS, RA PALS, and PACS before and after the race while LA PACS significantly increased after the race, as compared with baseline data (*p* < 0.001), demonstrating that biatrial reservoir and contractile function were not affected by the race. Therefore, our study supports the hypothesis that, in the absence of structural heart disease, an ultra-endurance race cannot influence biatrial function, and the cause of eventual supraventricular arrhythmias occurring in master athletes should be identified in triggers and substrate that differ from biatrial remodelling.

### 4.3. Electrical Function

Intensive training can induce acute electrocardiographic changes, which may reflect the physiological cardiac adaption to exercise and should not be considered as pathological findings [19]. In this study, similarly to previous findings [38], we found an increase in P voltage and more athletes fulfilling the criteria for RAE after the race, with no differences in the percentage of athletes with LAE. Similarly, Lord et al. investigated the right heart of 30 athletes running a 100-mile endurance race using ECG. The authors found a significant increase in P wave amplitude (28%) after the race, providing evidence that ultra-endurance races may induce transient changes in resting ECG, as demonstrated by other authors [38,39].

Marathon runners may exhibit higher atrial ectopy than nonmarathon runners, with premature atrial complexes increasing with lifetime training hours and the number of marathons participated in [40,41,42]. However, other studies did not confirm these results, in which former professional cyclists (mean age 66 ± 7 years) did not show a difference in the number of isolated atrial premature complexes or supraventricular tachycardias in the ambulatory 24-h ECG monitoring as compared with age-matched sedentary controls [43]. Similarly, Cipriani et al. demonstrated that middle-aged endurance athletes did not show a higher burden of PABs during 24-h ECG monitoring than sedentary controls [44]. Age, but not intensity and duration of sports activity, predicted a higher burden of PABs among healthy athletes [44]. Our study demonstrated that isolated PABs and more complex supraventricular arrhythmias are not uncommon in non-professional master athletes at rest. Notably, the prevalence of PABs and more complex supraventricular arrhythmias recorded during exercise was relatively low.

### 4.4. Implications for AF Pathophysiology in the Athlete

It has been hypothesised that the transient rise in cardiac and inflammatory biomarkers, acute inflammation, increased atrial wall tension, and acute atrial dysfunction may result in atrial damage after years of intense physical exercise, generating a substrate for atrial arrhythmias [16,17]. However, the relationship between AF and endurance training has been questioned [11] and the pathophysiological mechanisms responsible for developing AF in athletes remains unclear. Some studies demonstrated that an enlarged LA volume was associated with a higher risk of AF in endurance athletes [45]. However, LA size is known to be insufficient to provide mechanistic information about the left atrium itself, and peculiarities of LA remodelling in the context of the athlete’s heart go beyond the mere dimensional volumetric increase, including physiological changes in atrial function [11]. In a study conducted on professional soccer players, athletes exhibited significant differences in LA function, compared with controls, with normal reservoir function (i.e., PALS) and reduced LA active contribution (i.e., PACS) to LV diastolic filling at rest [34]. This finding was accompanied by supernormal diastolic function and was related to a shift of LV filling toward early diastole in the athlete’s heart [34]. Applying stiffness indexes to the athlete’s heart (a combination of the E/e’ ratio and PALS), it was demonstrated that, despite a greater LA size, athletes have a lower stiffness index compared to control subjects, demonstrating preserved compliance [26]. STE has been used in athletes to characterise RA deformation, confirming that in athletes, despite enlargement of the right atrium, functional properties remain normal, contributing through atrioventricular coupling to preload increase and stroke volume augmentation [46]. Notably, in our study, the athlete who developed AF during the race showed normal pre-race biatrial dimensions and preserved biatrial PALS and PACS. In agreement with our results, in a systematic investigation of more than 1700 elite athletes (24 ± 6 years, range, 11 to 56 years), examined during a 3-year period, AF (and other supraventricular tachyarrhythmias) was distinctly uncommon at study entry (<1%) and occurred with a similar frequency in athletes with enlarged and with normal LA size [27]. LA remodelling associated with intensive exercise and chronic athletic conditioning did not predispose *per se* to supraventricular tachyarrhythmias [27].

In conclusion, despite exercise-induced morphological atrial remodelling, the atrial function is normal in competitive athletes and should not be identified as a substrate for the occurrence of supraventricular arrhythmias, which conversely may be related to triggers, such as the influence of the autonomic nervous system, increased vagal tone, and changes in electrolytes [11].

### 4.5. Limitations

This study has some limitations. First, the data were collected before and after the race, immediately at the finish line. The lack of a post-race analysis and a follow up after the race represents a limitation of this study. However, atrial dysfunction was not found in this study, and, consequently, the evaluation of normalisation of a pathological finding was not needed. The evaluation of the chronic effects of ultra-endurance exercise represents a relevant topic, although it was beyond the scope of the study. Future research is needed to clarify the long-term sequelae of endurance exercise in master athletes, investigating the effects of different competitions and the impact of age and years of sports practice.

The results of our study may be related to the type of race and this specific population. Therefore, while these data demonstrate the absence of detrimental effects on atrial function in non-professional master athletes, the present results cannot be generalised to professional endurance athletes or athletes running different races (i.e., ironman). Finally, the athletes enrolled in this study were all Caucasians, and the results cannot be generalised to athletes of a different ethnicity.

## 5. Conclusions

In this study, we analysed the acute exercise-induced effects of an ultramarathon race on atrial remodelling and supraventricular arrhythmias in a population of non-professional master athletes. We demonstrated that, despite the presence of supraventricular arrhythmias, particularly at rest, acute exercise-induced atrial dysfunction was not detected, suggesting the lack of a role of biatrial functional remodelling as a substrate for the occurrence of supraventricular arrhythmias in this specific population and contrasting the hypothesis of an acute mechanical biatrial dysfunction induced by ultra-endurance exercise.

## Figures and Tables

**Figure 1 jcm-11-00528-f001:**
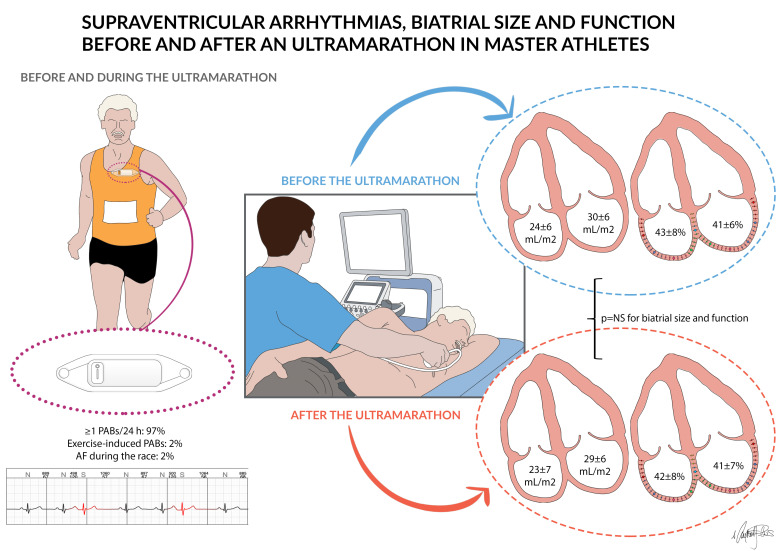
The figure illustrates the main findings of the study: data on supraventricular arrhythmias, atrial size, and function before and after the ultramarathon are shown.

**Table 1 jcm-11-00528-t001:** Demographic characteristics of the study population.

Variables	
Age, years	47.9 ± 7.8
Males, *n* (%)	47 (69)
Hours of Training per Week	6.7 ± 4.4
Years of Training	11.9 ± 8.2
Duration of Marathon Race, Hours	5.3 ± 7.6
Family History for CAD, *n* (%)	10 (15)
Family History for SCD, *n* (%)	0 (0)
Height, cm	173 ± 8
Weight, kg	71 ± 11
BSA, m^2^	1.8 ± 0.17

CAD, coronary artery disease; SCD, sudden cardiac death; BSA, body surface area.

**Table 2 jcm-11-00528-t002:** Twelve-lead resting ECG data of atrial electrical remodelling, collected before and after the race, in master athletes running an ultramarathon.

Variables	Pre-Race	Post-Race	*p* Value
Resting heart rate, bpm	61 ± 8	85 ± 14	<0.0001
PR interval, ms	157 ± 24	156 ± 20	0.49
P wave voltage, mV	0.16 ± 0.05	0.20 ± 0.0.6	<0.0001
P wave duration, ms	98 ± 17	103 ± 16	0.094
RA enlargement, *n* (%)	5 (7.4)	15 (22)	<0.0001
LA enlargement, *n* (%)	22 (32.4)	23 (33.8)	0.66

RA, right atrial; LA, left atrial. *n* (%): number of athletes, %: percentage of athletes.

**Table 3 jcm-11-00528-t003:** Ambulatory ECG monitoring data obtained the day before the race and during the entire duration of the race. Data obtained at rest and during the race were reported separately.

Variables	
Min HR, bpm	41 (38–45)
Max HR, bpm	172 (161–182)
Max Theoretical HR, %	98 (91–105)
Number of PABs/24 h, *n* ^	7 (3–19)
≥1 PAB(s)/24 h, *n* (%)	57 (97%)
≥100 PABs/24 h, *n* (%)	5 (8%)
≥1 PAB(s) during the Race, *n* (%)	13 (22%)
Exercise-Induced PABs *, *n* (%)	1 (2%)
≥1 Couplet(s), *n* (%)	12 (21%)
≥1 Couplet(s) during the Race, *n* (%)	2 (4%)
≥1 Triplet(s), *n* (%)	10 (17%)
≥1 Triplet(s) during the Race, *n (%)*	1 (2%)
≥1 Non-Sustained SVT(s), *n (%)*	11 (19%)
≥1 Non-Sustained SVT(s) during the Race, *n (%)*	0

HR: heart rate; PAB: premature atrial beat; SVT: supraventricular tachycardia. ^ *n* in this case means median number of PABs in one day. * Occurrence of PABs only during exercise or an increase in the burden of PABs/hour. *n*: number of athletes.

**Table 4 jcm-11-00528-t004:** Echocardiographic characteristics of the left and right atrium observed before and after the race in master athletes running an ultramarathon.

Echocardiographic Variables	Pre-Race	Post-Race	*p* Values
Left atrium			
LA volume, mL	54.3 ± 11.5	52.2 ± 12.1	0.15
LA volume index, mL/m^2^	30.0 ± 5.5	28.6 ± 6.2	0.16
LA PALS, %	40.6 ± 6.1	41.1 ± 7.1	0.46
LA PACS, %	18.3 ± 4.4	20.7 ± 5.5	0.001
Right atrium			
RA volume, mL	43.6 ± 11.3	42.1 ± 12.3	0.25
RA volume index, mL/m^2^	23.8 ± 6.0	23.1 ± 6.9	0.31
RA PALS, %	43.4 ± 7.8	41.7 ± 7.8	0.057
RA PACS, %	19.0 ± 5.0	20.9 ± 7.5	0.076

LA, left atrial; PALS, peak atrial longitudinal strain; PACS, peak atrial contraction strain; RA, right atrial.

## Data Availability

Data are available upon reasonable request.

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
