# Peer review of "The Acute Effects of an Ultramarathon on Atrial Function and Supraventricular Arrhythmias in Master Athletes"

_jcm, 2022, doi:10.3390/jcm11030528_

Round 1

Reviewer 1 Report

In master athletes running an ultramarathon, compared with baseline, P wave voltage was higher after the race (p<0.0001), and more athletes developed ECG criteria for right atrial enlargement (p<0.0001). However, exercise-induced atrial dysfunction was not detected and exercise-induced supraventricular arrhythmias were uncommon.

This paper is interesting.

  1. Although the change of echocardiography was not observed, ECG change was observed after exercise. It would be interesting to discuss the electrical change after exercise.

Author Response

See file attached

Reviewer 2 Report

In this study, Dr. Cavigli and colleagues investigated the acute effects of ultramarathon on atrial function and supraventricular arrhythmias in master athletes. Overall, the manuscript was nicely written and quite clear. It is also relevant to discuss. Nonetheless, I have some suggestions and comments to address:

  • Line 24: "Most of the athletes had ≥1 premature atrial beats (PAB) during the 24-hour monitoring" please add the percentage or the absolute number to increase clarity. 
  • Regarding this statement "In contrast, exercise-induced PABs, triplets and NSSVT were rare." were these parameters assessed during exercise only or also post-exercise? Because some abnormalities can occur a few hours after the race and yet it is still exercise induced. I think it is better to use a terminology "acute exercise-induced XXX" to make it clearer that the authors only focus on the abnormalities found during exercise. Consider revising.
  • Since the conclusion of the abstract was about "exercise-induced XXX", I think it is very important to describe what was the definition of "exercise-induced" in this study and how long was the observation time (during and after exercise) so that a phenomenon can still be considered as "exercise-induced"?
  • In the Methods section, please specify the ECG criteria to define RAE and LAE used in this study.
  • In Table 3, "≥1 PAB(s)/24 hours, n/%" what is the meaning of this "n"? number of athletes? or number of PABs? Please clarify also the rest in that table (please also check other tables for a similar ambiguity). 
  • Also, I am confused, what is the difference between "≥1 PAB(s) during the race, n/%" and "Exercise-induced PABs*, n/%"? The former is also exercise-induced PABs, isn't it?
  • I am not sure what the significance of the "correlation analysis" section. Is it needed in this study? If so, make it clearer. I don't think it is even discussed in the discussion. 
  • Line 289: "...(16, 17) may result..." Please remove the citations. They are already mentioned at the end of the sentence.
  • Regarding this statement "Notably, in our study, the athlete who developed AF during the race showed normal pre-race biatrial dimensions and preserved biatrial PALS and PACS.", I think it would be more informative and useful if the authors compared the ECG, echo, STE and other data of this patient with the rest of the athletes. The authors could make a table comparing the two. 
  • It would be interesting to have some examples / images of the abnormal ECG recordings or echo / STE parameters. Consider providing, including the one from the AF case. 
  • "... and refusing the hypothesis of an acute mechanical biatrial dysfunction induced by ultra-endurance exercise" I think it is better to tone down "refusing" to "contrasting". 
  • I couldn't find Figure 1 (or any figure in general) but there is a legend for this figure after the references. Please correct this error. 

Author Response

See file attached

Round 2

Reviewer 2 Report

Thank you for the clear and responsive reply. I think everything looks good so I have no further remarks.